# Immunohistochemical Analysis of Mastocyte Inflammation: A Comparative Study of COPD Associated with Tobacco Smoking and Wood Smoke Exposure

**DOI:** 10.3390/biomedicines13071593

**Published:** 2025-06-30

**Authors:** Robinson Robles-Hernández, Rosa María Rivera, Marcos Páramo-Pérez, Dulce Mariana Quiroz-Camacho, Gustavo I. Centeno-Saenz, Alan Bedolla-Tinoco, María C. Maya-García, Rogelio Pérez-Padilla

**Affiliations:** 1Departamento de Investigación en Tabaquismo y EPOC, Instituto Nacional de Enfermedades Respiratorias, Mexico City 14080, Mexico; robinsonrobher@gmail.com (R.R.-H.); igcenteno89@gmail.com (G.I.C.-S.);; 2Departamento de Patología, Instituto Nacional de Enfermedades Respiratorias, Mexico City 14080, Mexico; rosa.rivera48@yahoo.com.mx (R.M.R.); ecludqc1@gmail.com (D.M.Q.-C.)

**Keywords:** mast cells, airway diseases, COPD biomass, wood smoke, tobacco smoking, phenotypic differences, immunohistochemistry

## Abstract

**Background:** Chronic Obstructive Pulmonary Disease (COPD) exhibits some phenotypic differences between patients with biomass smoke inhalation (COPD-B) and tobacco smoking (COPD-T). COPD-B is characterized by less emphysema but more airway disease and vascular pulmonary remodeling, which are related to mast cells in lung tissues in COPD-T. Our objective was to describe the differences between the number of mast cells in COPD-B and COPD-T patients. **Methods**: A cross-sectional study was conducted on lung tissue resections for suspected cancer obtained between 2014 and 2021 from patients with documented COPD due to wood smoke or tobacco exposure. Histological samples were analyzed for mast cell count, CD34+ expression, and structural changes in lung tissue and pulmonary circulation. **Results**: A total of 20 histological samples were analyzed, with significant differences found in mast cell count [median 8 (p25-75, 5–11) vs. 2 (p75-25, 0–6), *p* = 0.016] and severe peribronchiolar fibrosis (60% vs. 10%, *p* = 0.04) between COPD-B and COPD-T patients. A positive correlation [Spearman rho = 0.879 (95% CI 0.71–0.96), *p* < 0.001] was observed between mast cell count and a gradual increase in pulmonary artery diameter. **Conclusions**: These preliminary findings suggest histological differences and the presence of mast cells between COPD-B and COPD-T, which should be confirmed in a larger number of samples and patients.

## 1. Introduction

Chronic obstructive pulmonary disease (COPD) poses a significant health burden and is the third cause of death in the world [1]. In Mexico, it is the principal cause of respiratory death [2]. Several international meta-analyses estimate that approximately 15–20% of COPD cases in developing countries are attributed to biomass smoke exposure, compared to 55–60% attributable to smoking [3]. In Mexico, these proportions shift, with one study reporting that 23% of COPD patients are non-smokers [4].

This disease is characterized by an inflammatory process induced by noxious particles or gases, which can activate a cascade of inflammatory reactions, leading to tissue destruction in the airways [5,6]. Most evidence on mechanisms of damage induced by inhaled agents is based on COPD studies due to tobacco smoke (COPD-T); however, phenotypic differences have been documented between exposure to biomass smoke (COPD-B) and COPD-T [7,8].

Some studies describe a Th2 inflammatory phenotype in COPD-B, characterized by higher levels of sputum and serum eosinophils, or fractional exhaled nitric oxide (FeNO) [3,4,7]. Women exposed to biomass smoke present more peribronchiolar fibrosis and small airway disease [9,10,11]. In addition to these differences, greater pulmonary circulation and pulmonary hypertension remodeling have been documented [12,13], consistent with findings in various histological studies [9,12,14]. These differences suggest the existence of two different COPD phenotypes, one related to tobacco smoke and the other to biomass.

The role of mast cells in lung tissue has been studied in patients with COPD-T. Infiltration of the small airways by mast cells has been reported, along with changes in the connective tissue/mucosa balance, which are associated with airway hyperreactivity [15]. An increase in the density of perivascular mast cells has also been observed, correlating with increased levels of vascular endothelial growth factor (VEGF) and the development of pulmonary hypertension [16,17].

At the time of this review, we did not find any reports on the role of mast cells in COPD-B; therefore, our objective was to explore the differences in the number of mast cells in lung tissue between COPD-B and COPD-T, and to characterize histological differences in the airways and pulmonary vasculature.

## 2. Materials and Methods

### 2.1. Design and Selection of Histological Samples

This study is a cross-sectional investigation conducted at the Instituto Nacional de Enfermedades Respiratorias, a referral center in Mexico for the treatment of respiratory diseases. This study was submitted to and approved by the local ethics committee under number C50-22. A series of surgical resections of lung parenchyma obtained between 2014 and 2021 and were reviewed by a pathologist. The main indications for resections were in patients with a primary nodule suspected of cancer. A total of 13 out of the 20 patients had a diagnosis of cancer based on nodules or tumors smaller than 3 cm. Additionally, segmental or lobar resections were performed secondarily for hemoptysis or nodules with no malignancy, often accompanied by nonspecific inflammation.

The selection criteria for the analysis of the samples were patients over 40 years of age with a diagnosis of COPD due to wood smoke or to tobacco smoking in the clinical record, or those who had a cumulative exposure to wood smoke greater than 200 hour-years [18], or tobacco consumption >10 pack-years [19]. Samples from patients who received inhaled or systemic steroids within 30 days of the biopsy, patients with records of active infectious processes corroborated by cultures or visualization of the parenchyma, or those suspected of having interstitial lung disease were excluded. In parallel, the clinical information of the patients was reviewed, including demographic data, medical history, and respiratory function, particularly spirometry with bronchodilator (FEV_1_/FVC, FEV_1_, and FVC), as well as tomographic measurements of the pulmonary artery diameter. 

### 2.2. Tissue Procurement and Processing

During the histological review of the sections stained with H-E, it was ensured that they presented sufficient and optimal material for evaluation and that the parenchyma was free of malignant cells; two pathologists reviewed all slides.

Immunohistochemistry (IHC) was performed on the paraffin blocks of the selected cases using 3 µm histological sections, with mouse monoclonal anti-human chymase antibodies (Mouse Monoclonal Mast Cell Chymase antibody, clone Cf1, ab2377; Abcam plc, Cambridge, UK; dilution 1:100) from a murine source. To determine the positivity of the samples, they were contrasted with the tissue controls suggested by the manufacturer (esophagus and brain).

### 2.3. Definition of Lung Regions and Morphometric Analysis

One to five peripheral airways were evaluated in 10 fields at 40× for each case. The variables assessed in the respiratory tract were as follows: (1) epithelium, (2) peribronchiolar subepithelial area, and (3) smooth muscle.

Peripheral airways were defined as airways with a circumferential size of ≤6 mm to >2 mm, and small airways were described as smaller than ≤2 mm.

### 2.4. Measurements by Pathologists

The IHC reactions were analyzed at 40× magnification in 10 high-power fields using a digital pathology scanner (Aperio CS2; Leica Biosystems Imaging, Inc., Vista, CA, USA for Digital Preparations), creating files for each selected case and corresponding antibody. The first pathologist selected each field, and the second pathologist evaluated the same fields, but both were blind to the reports and counts of the other.

The histological samples were analyzed semi-quantitatively, defining the intensity of changes using crosses on a visual scale from 0 to 3, according to the following grades: 0 = absent, 1+ = Mild, 2+ = moderate, 3+ = intense. The changes identified in patients with COPD-T and COPD-B were reported as follows:(1)Inflammation: inflammatory infiltrate of lymphocytes in the airway wall.(2)Peribronchiolar fibrosis: the amount of fibroconnective tissue deposited in the wall of the airway(3)Goblet cell metaplasia: replacement of a mature epithelial cell by another mature cell different from the original(4)Intensity of the affinity of anti-CD34+ for the microvascular endothelium.(5)A single pathologist counted mast cells marked by CC1 in the peribronchial region of the small airway. The pathologist performed two counts, separated by a 14-days, and averaged them to produce a repeatable measurement.

### 2.5. Statistical Analysis

Differences between groups were analyzed using Chi-Square or Mann–Whitney U tests. Paired analyses within groups were performed using Wilcoxon rank sum tests. Correlations between cell concentrations and lung function were assessed using Spearman’s rank correlation coefficient. A 95% confidence interval for rho was estimated via a non-parametric bootstrap, using the 2.5th and 97.5th percentiles of the bootstrap distribution.

Two-tailed *p* values < 0.05 were considered significant. To analyze the repeatability of the semiquantitative scale, agreement between pathologists was calculated using the kappa coefficient for the categorical measurements, and a Band–Altmann analysis was performed for the mast cell count.

Statistical analysis was performed using Stata 17.1 software.

## 3. Results

A total of 118 histological samples were taken from patients exposed to wood smoke or tobacco smoke. Ninety-two samples were discarded due to insufficient lung tissue, bronchoscopy biopsy, tumor infiltration, or lack of data on granulomatous inflammation. Two samples were excluded because the patients had an insufficient smoke exposure, and other two because had recently received antibiotics.

Among the main demographic characteristics, women predominated in the COPD-B group. No differences were observed in comorbidities, and the leading cause of lung resection was confirmed lung cancer. There were no differences in lung function or peripheral eosinophils (see Table 1).

Histological evaluations showed high concordance between the two pathologists. For categorical variables on the semiquantitative scale, kappa coefficients ranged from 0.72 to 0.92 (depending on the characteristic evaluated), with observed agreement percentages >80%. (Table 2). For quantitative cellular measurements, 20 observations without missing data were reported. First measurement (M1) was 6.1 (95% CI 2.3–9.9) and the mean for measurement 2 (M2) was 6 (95% CI 2–9.2). Bland–Altman analysis showed a mean bias of 0.1. The limits of agreement were −4.73 cells for the lower limit and 4.36 cells for the upper limit. There was one case (5.00%) that exceeded the upper limit; Lin’s coefficient of agreement was 0.96 (Appendix A). Taken together, these concordance metrics demonstrate that our histological evaluation method is both repeatable and reproducible, despite incorporating semi-quantitative methods.

Patients exposed to biomass had significantly more peribronchiolar fibrosis (Table 3) and a higher total mast cell count than tobacco smokers (*p* = 0.0204, Figure 1). The mast cells and CD34+ intensity were identified in vascular endothelium both in COPD-B and COPD-T (Figure 2).

Figure 3 illustrates a positive correlation between the number of mast cells and the gradual increase in the diameter of the pulmonary artery, as measured by tomography.

Figure 4 highlights the greater number of mast cells and increased intensity of microvascular endothelial expression of the anti-CD34+ antibody, as well as an increased number of mast cells associated with smooth muscle hyperplasia, with statistically significant differences (*p* < 0.05) in these characteristics.

## 4. Discussion

This pilot analysis provides descriptive and hypothesis-generating evidence on possible tissue differences between COPD-B and COPD-T, identifying a difference in total mast cell counts in patients with COPD-B, as well as in the expression of CD34+ in the vascular endothelium and peribronchiolar fibrosis. Furthermore, a positive correlation was found between the diameter of the pulmonary artery and the total number of mast cells. In addition, there was a trend toward greater smooth muscle hyperplasia, increased intensity of CD34+ in the endothelium of the pulmonary circulation, and a higher number of mast cells in the samples from COPD-B.

These findings enable us to explore and hypothesize that mast cells may contribute to understanding specific characteristics of patients with COPD-B, including small airway disease-related inflammation [10] and eosinophilic inflammation resistant to steroid treatment, as shown by Salvi et al. [7]. For example, Amir Soltani et al. described hyperreactivity to the airways and the presence of mast cells in the airways of patients with COPD [15], in addition to less severe obstruction in patients with COPD-T who have a greater number of mast cells [16]. Both phenomena have also been previously reported in patients exposed to wood smoke [20,21,22]. Smooth muscle remodeling and hyperplasia were observed in COPD-B; however, this association warrants further investigation in larger and more detailed studies. The presence of mast cells may be mediated by an intricate stimulation of pro-Th2 and Th17 cytokines, as well as various epigenetic mechanisms described in some COPD-B patients [23]. Mast cells have been described as a key cell that can accompany a Th2 response, respond to steroids, or be associated with peripheral eosinophilia [24,25]; however, another phenotype involving migration to the submucosal and connective tissue compartments has also been described. This is driven by non-Th2-IgE interleukins [26], especially TNFa, IL-8, IL-1b, IL17R, and inflammasome activation [27,28], which in turn have been identified as predominant cytokines in COPD-B [23,29,30].

We observed a modest positive correlation between mast cell counts and CT-derived pulmonary artery diameter, as well as a higher CD34^+^ staining intensity. These associations are compatible with the hypothesis that mast cells may participate in pulmonary vascular remodeling; multiple reports have suggested that mast cells may be involved in vascular remodeling and angiogenesis and are associated with pulmonary hypertension in patients with COPD [17]. In this sense, local recruitment in different compartments and expressions of inflammatory phenotypes in tissues influenced by cytokines of adaptive immunity are of particular importance [17], as they prolong mast cell survival and localize them mainly in the small airways and connective tissues [31]. Mast cells regulate angiogenesis through the production of VEGF, angiotensin II, and the release of proangiogenic proteases [17,31], but the current data are insufficient to draw causal inferences and should be confirmed in larger, mechanistic studies.

The CD34+ molecule has been described as having greater expression in the endothelium of patients with pulmonary hypertension [20]; in addition, it has also been linked to mast cell migration [32]. What makes mast cells particularly interesting is that they degranulate in hypoxic conditions, which in turn causes an imbalance in reactive oxygen species (ROS) and nitric oxide [33,34]. As described in different cohorts, patients with COPD-B show this characteristic [35,36].

Other pathways may also contribute. In tobacco-related COPD (COPD-T), several studies have reported mast-cell activation with chymase-1 release, an event that could favor inflammation, macrophage influx, and airway remodeling or emphysema [37,38]. Biomass smoke exposure leads to increased IL-17a levels, which correlate with small airway disease and remodeling. The nuclear receptor NR1D1 modulates this process [39,40]. While mast cells are likely involved in both phenotypes, the downstream pathways and predominant inflammatory mediators differ, with chymase-1 and IL-33 being more central in COPD-T and IL-17a and NR1D1 in COPD-B. Adipokine imbalance has also been proposed as a contributing factor: elevated leptin and insulin resistance may promote Th1/Th17 polarization and oxidative stress, mechanisms that have been correlated with biomass-associated COPD [41].

### 4.1. Limitations

Our sample size was small, which prevented multivariate analyses adjusting for potential confounders. Furthermore, a healthy control group was not included, as obtaining normal lung tissue in our clinical setting represents an ethical and practical challenge. This precludes comparisons with baseline reference values in healthy tissue. Furthermore, specific angiogenic markers (e.g., VEGF) were not evaluated in immunohistochemical staining, which would have provided additional information on neovascularization processes. The immunohistochemical patterns recommended by the manufacturer were used as references to determine the staining as positive. Most of the tissues obtained came from patients with suspected cancer. Although diseased tissue was excluded and the absence of malignant cells or associated changes was confirmed, the possibility that the tumor microenvironment influenced the findings cannot be ruled out.

Another limitation is that mast cell counting was performed manually in selected microscopic fields, introducing the possibility of interobserver variability; some residual recall bias in intraobserver counts cannot be completely ruled out. Therefore, intraobserver statistics should be interpreted as estimates of the upper limit of reproducibility. Automated image analysis tools were unavailable due to resource limitations, which would have enhanced the objectivity of quantification.

Our data suggest that mast cells may contribute to peribronchiolar fibrosis and vascular remodeling in COPD-B. While previous studies have linked mast cell mediators to angiogenesis, Th2/Th17 polarization, and hypoxia-induced degranulation, these mechanisms remain speculative in our data and warrant specific functional investigation.

### 4.2. Future Directions

This pilot study provides preliminary mechanistic information on the histopathological differences between B- and T-COPD. More robust study designs will be essential for determining causality and the temporal relationship between exposure, inflammatory mast cell infiltration, and tissue remodeling. Automated measurement could refine mast cell quantification and immunohistochemical measurements specific for vascular remodeling, such as VEGF, as well as measurements of osteopontin, eosinophilic infiltration, and others, which could elucidate pulmonary hypertension (PH) and inflammatory mechanisms.

## 5. Conclusions

This pilot study identifies histological differences between COPD-B and COPD-T. COPD-B patients may exhibit increased mast cell infiltration, peribronchiolar fibrosis, and CD34+ expression in the vascular endothelium compared with COPD-T patients. We also observed a positive association between mast cell count and pulmonary artery diameter. However, the small sample size limits the robustness and generalizability of these results. The observations presented here should be considered preliminary, pending validation in multicenter studies with healthy controls, automated quantification methods, and specific mechanistic analyses.

## Figures and Tables

**Figure 1 biomedicines-13-01593-f001:**
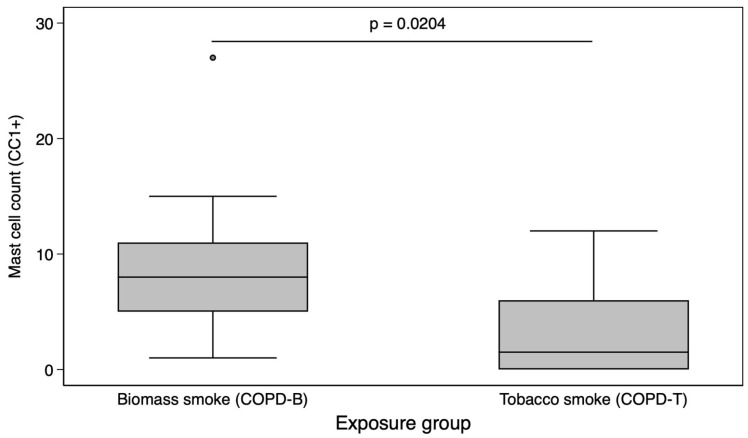
Comparison of mast cell counts by exposure type (biomass vs. tobacco). Boxplot comparing mast cell counts (CC1+ positive cells per high-power field) between COPD patients exposed to biomass smoke (COPD-B, left) and those exposed to tobacco smoke (COPD-T, right). Data represents medians and interquartile ranges. A significantly higher mast cell count was found in the COPD-B group (Mann–Whitney U test, *p* = 0.0204).

**Figure 2 biomedicines-13-01593-f002:**
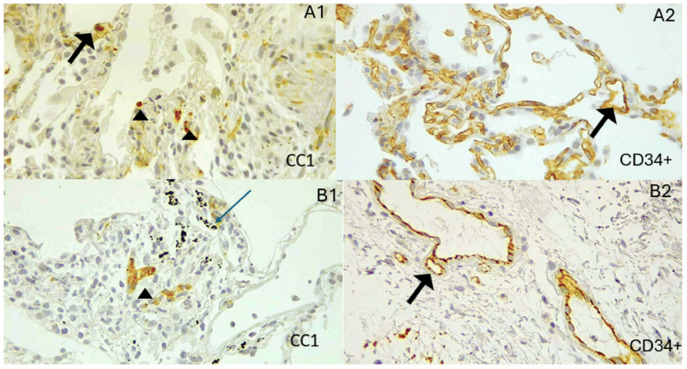
Representation of mast cell count (CC1+) and CD34+ intensity in vascular endothelium in COPD-B and COPD-T. (**A1**,**A2**) Biopsy from a patient with COPD-B. (**A1**) Mast cell clusters (arrowhead) are distributed in the submucosal regions and vascular intima (marked by arrows). (**A2**) The CD34+ marker shows significant intensity in the vascular endothelium (arrows). (**B1**,**B2**) Biopsy from a patient with COPD-T. (**B1**) The mast cell marker shows a cluster of mast cells in the submucosa (arrowhead) and alveolar anthracosis (thin arrow, blue). (**B2**) The endothelial marker CD34+ shows less intensity. Images were obtained at 40× magnification. Quantification was based on ten high-power fields per sample, in regions selected for preserved alveolar structure.

**Figure 3 biomedicines-13-01593-f003:**
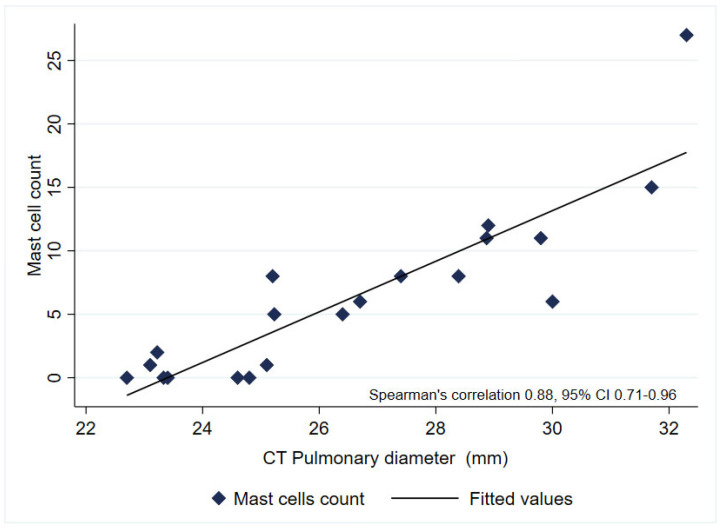
Correlation between pulmonary artery diameter and mast cell count in COPD patients. Scatterplot showing the association between pulmonary artery diameter measured by chest CT (*x*-axis) and the number of mast cells identified by immunohistochemistry (CC1+ staining) in lung tissue (y-axis). Each point represents one patient sample. The fitted line represents linear regression. A strong positive correlation was observed (Spearman’s rho = 0.879, 95% CI 0.71–0.96, *p* < 0.001).

**Figure 4 biomedicines-13-01593-f004:**
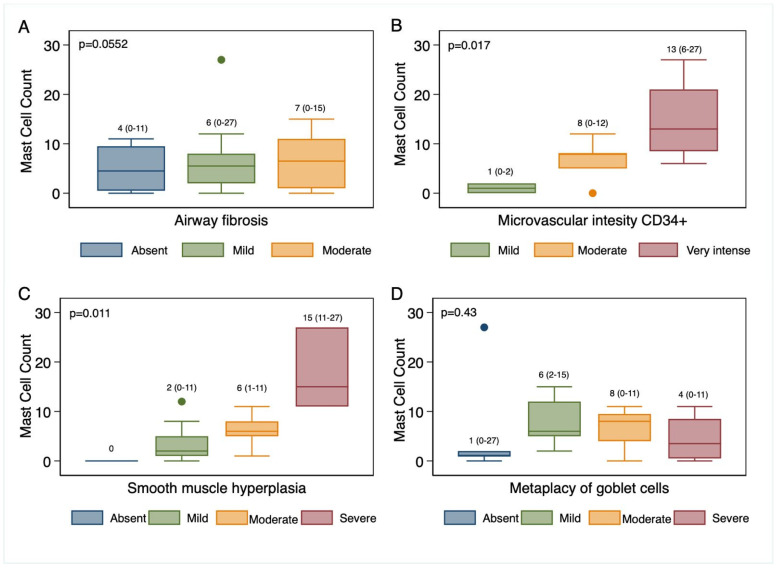
Mast cell total count and distribution of four histologic features. Non-parametric tests, such as the Kruskal–Wallis test, were used to compare medians among independent groups. (**A**): airway fibrosis categories. (**B**): Microvascular intensity. (**C**): Smooth muscle hyperplasia. (**D**): Metaplasia of goblet cells.

**Table 1 biomedicines-13-01593-t001:** Demographic and clinical characteristics.

	COPD–Biomass (*n* = 10)	COPD–Tobacco (*n* = 10)	*p* Value
Female, n (%) **^¥^**	8 (80)	1 (10)	0.003
Age, median (p25-75) *	63 (46–78)	68.5 (55–76)	0.382
Pack-years, median (p25-75) *	-	45 (37.5–68)	-
Hour–years of biomass exposure, median (p25-75)	238 (180–308)	-	-
Diabetes, n (%) **^¥^**	4 (40)	2 (20)	0.314
Previous COPD diagnosis, n (%)	7 (70)	9 (90)	0.291
Arterial hypertension, n (%) **^¥^**	5 (50)	5 (50)	0.672
Ischemic heart disease, n (%) **^¥^**	1 (10)	5 (50)	0.070
Lung cancer biopsy, n (%) **^¥^**	7 (70)	6 (60)	0.5
Non specific inflammation biopsy, n (%) **^¥^**	3 (30)	4 (40)	
FEV_1_/FVC, median (p25-75) *	0.53 (0.47–0.63)	0.61 (0.55–0.87)	0.158
FEV_1_ (L), median (p25-75) *	1.02 (0.97–1.31)	1.8 (0.88–2.58)	0.408
FEV_1_ %, median (p25-75) *	60 (49–75)	65 (39–78)	0.796
FVC (L), median (p25-75) *	2.01 (1.9–2.1)	2.66 (1.4–4.2)	0.814
FVC %, median (p25-75) *	75 (64–89)	78.5 (45–101)	0.10
Pulmonary artery diameter (mm), median (p25-75) *	27.9 (25.2–29.8)	24.9 (23.3–29.4)	0.143

^¥^ Fisher’s test; * U Mann–Whitney test. Abbreviations: COPD, chronic obstructive pulmonary disease; p25-75, 25th to 75th percentile; FEV_1_, Forced Expiratory Volume in the first second; FVC, Forced Vital Capacity.

**Table 2 biomedicines-13-01593-t002:** Semiquantitative scale, Cohen’s kappa concordance among pathologists.

	Kappa Coefficient	Agreement (%)
Small airway inflammation *	0.869	90
Peribronchiolar fibrosis *	0.771	85
Pigment deposit *	0.921	95
Smooth muscle hyperplasia *	0.773	85
Goblet cell metaplasia *	0.855	95
Microvascular intensity of CD 34+ *	0.719	80

* Statistical significance *p* > 0.001.

**Table 3 biomedicines-13-01593-t003:** Histological features for the semiquantitative report and quantitative cell count for mast cells.

	COPD–Biomass (n = 10)	COPD–Tobacco (n = 10)
Small airway inflammation ^∂^		
Mild +	5	4
Moderate ++	4	5
Severe +++	1	1
Peribronchiolar fibrosis		
Absent −	0	3
Mild+	4	6
Moderate++ ^β^	6	1
Pigment deposit ^∂^		
Mild +	6	5
Moderate ++	3	3
Severe +++	1	2
Smooth muscle hyperplasia ^∂^		
Absent −	1	3
Mild +	0	2
Moderate ++	6	4
Severe +++	3	1
Goblet cell metaplasia ^∂^		
Absent −	1	3
Mild +	0	2
Moderate ++	6	4
Severe +++	3	1
Microvascular intensity of CD34+ ^∂^		
Absent −	0	2
Mild +	2	4
Moderate ++	4	3
Severe +++	4	1

Statistical analysis: for categorical variables, Fisher’s test was applied: **^β^** *p* = 0.029; **^∂^** *p* > 0.05 for continuous variables. The intensity of each qualitative characteristic evaluated by the pathologists was expressed using a visual analog scale: “−” indicating the absence of the finding, “+” for mild intensity, “++” for moderate, and “+++” for intense.

## Data Availability

The datasets used and/or analyzed during the current study are available from the corresponding author upon reasonable request.

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
