# Peer review of "Immunohistochemical Analysis of Mastocyte Inflammation: A Comparative Study of COPD Associated with Tobacco Smoking and Wood Smoke Exposure"

_biomedicines, 2025, doi:10.3390/biomedicines13071593_

Round 1
Reviewer 1 Report
Comments and Suggestions for Authors
The article has successfully investigated a comparative study in COPD associated with tobacco smoking and wood smoke exposure. The research content is relatively systematic, and logically clear. However, there are still areas in the article that need improvement.
- The study in this article is a cross-sectional design with a small sample size (only 20 cases) and lacks a healthy control group. These factors may limit the generalizability and interpretability of the findings. It is recommended to thoroughly discuss these limitations in the discussion section and propose directions for future research.
- In lines 33-34, the authors mention that the characteristic feature of COPD is an inflammatory process caused by noxious particles or gases, but they do not provide an in-depth analysis of these particles or gases. It is recommended that the authors analyze airborne particulate matter and cite relevant studies to enhance the reliability of the article, such as Atmosphere, 2024, 15, 384, Adv. Mater., 2020, 32, 2002361, Nat. Commun., 2024, 15, 5116.
- The authors analyzed that the number of mast cells in the lung tissue of COPD-B patients was significantly higher than that of COPD-T patients, but did not provide specific mast cell distribution maps or related imaging data. The inclusion of relevant graphical data is recommended to facilitate more intuitive interpretation of the research findings.
- The authors reported a positive correlation between mast cell density and pulmonary artery diameter, yet fails to elucidate the potential mechanistic underpinnings of this association. It is recommended that the discussion section incorporate:
(1) An in-depth exploration of plausible biological mechanisms (for example, mast cell-derived vasoactive mediators promoting vascular remodeling).
(2) Systematic comparison with congruent findings in existing literature.
Author Response
We appreciate your concise and, above all, respectful comments, all with the goal of improving the research. We've adjusted below based on the points discussed.

Reviewer 2 Report
Comments and Suggestions for Authors
This manuscript examines histological differences in mast cell infiltration and vascular remodeling between biomass-related COPD (COPD-B) and tobacco-related COPD (COPD-T). The authors report higher mast cell counts, increased peribronchiolar fibrosis, and elevated CD34+ endothelial expression in COPD-B, suggesting a distinct pathophysiological mechanism involving mast cell–mediated inflammation. While the topic is relevant and underexplored, the study is weakened by methodological and interpretive limitations that undermine its scientific value and reproducibility.
A central issue is the lack of a clearly defined, mechanistically testable hypothesis. The rationale for emphasizing mast cells is underdeveloped, and the link between mast cell activation, CD34+ expression, and vascular changes remains speculative, lacking molecular or functional validation. The purpose and meaning of “LH” in line 44 also require clarification.
The small sample size (10 cases per group) limits statistical power, particularly given interindividual variability in histological features. The absence of healthy lung tissue as a control further hampers interpretation, as baseline levels of mast cells and vascular markers cannot be established. Although this limitation is acknowledged, it remains a significant weakness.
All samples were derived from patients undergoing resection for suspected malignancy. Although tumor-free tissue was used, potential confounding from tumor-adjacent inflammation or systemic disease was not addressed. This is particularly concerning in an immunohistochemical context.
Statistical reporting lacks rigor. While non-parametric tests are appropriate, no effect sizes or confidence intervals are provided. Semiquantitative grading scales are used without validation or detailed inter-rater reliability analysis. The reported correlation between mast cell density and pulmonary artery diameter lacks adjustment for confounding variables such as age or comorbidities.
Figures are inadequately annotated. Immunohistochemistry images are not supported by quantification, and figure legends do not explain region selection or analysis methods. Correlations in Figures 3 and 4 are presented without multivariate modeling, and the relationship between CD34+ staining and mast cells is implied but not directly analyzed.
Finally, the discussion overstates conclusions, linking mast cells to angiogenesis, Th2/Th17 pathways, and hypoxia mechanisms without direct evidence. While plausible, these associations are speculative and rely heavily on inference. The assertion that mast cells are "key" drivers of COPD-B pathology is not substantiated by the presented data and should be substantially tempered.
Although the writing is generally fluent, the manuscript would benefit from language tightening and improved clarity in terminology. For instance, terms like “mastocyte inflammation” and “inflammatory phenotypes” are inconsistently applied, and in some places, the language is redundant or vague.
Author Response

(The authors gave the same response as above.)

Reviewer 3 Report
Comments and Suggestions for Authors
Review report
I read this study with great interest. The authors address a clinically important and timely topic by exploring the pathological differences between two major etiologies of COPD: tobacco smoke (COPD-T) and biomass smoke (COPD-B) exposure. The present study seeks to address this gap by employing immunohistochemical methods to compare mast cell infiltration and related structural changes in lung tissue samples from patients with COPD-B and COPD-T.
I have only a few comments for further improvement:
- The Introduction and the Abstract could be strengthened by incorporating the latest epidemiological data on the global and national burden of COPD, including recent statistics on its incidence and prevalence in Mexico. Providing this context would help to highlight the clinical and public health significance of investigating the overlap between biomass and tobacco smoke exposure in COPD, thereby emphasizing the relevance of the present study.
- The Introduction would be greatly improved by including more specific numerical data regarding the incidence and prevalence of COPD related to both biomass smoke (COPD-B) and tobacco smoke (COPD-T). For example, citing recent estimates of the percentage of COPD cases attributed to biomass versus tobacco exposure globally and in Mexico would provide clearer context.
- The manuscript would benefit from citing recent articles in the Discussion sections, such as “Integrative Approach to Risk Factors in Simple Chronic Obstructive Airway Diseases of the Lung or Associated with Metabolic Syndrome Analysis and Prediction” and “Leptin and Insulin in COPD: Unveiling the Metabolic-Inflammatory Axis A Narrative Review,” to provide broader context on metabolic and inflammatory factors in COPD.
I also recommend including more numerical data on the global and national incidence and prevalence of COPD, as well as the proportion of cases attributable to biomass versus tobacco smoke, to better highlight the study’s clinical and public health significance.
- The Discussion section could be enhanced by comparing your findings to more recent literature on mast cell involvement and airway remodeling, exploring potential mechanisms for the observed differences between COPD-B and COPD-T, and discussing the clinical implications for targeted therapies and public health interventions in at-risk populations.
- A dedicated Limitations section should be included in the manuscript. This section should clearly acknowledge and discuss the study’s main constraints, such as the small sample size, the lack of a healthy control group. Addressing these limitations openly will enhance the transparency and scientific rigor of the article.
- Including a section on Future Directions would further strengthen the manuscript. This section could outline potential avenues for continued research, such as larger multicenter studies, longitudinal analyses to assess causality, the inclusion of healthy control tissues, and expanded biomarker profiling to further elucidate the mechanisms differentiating COPD-B and COPD-T.
Author Response

(The authors gave the same response as above.)

Round 2
Reviewer 2 Report
Comments and Suggestions for Authors
Although the authors have addressed many of the prior concerns and demonstrated a constructive attitude in revising the manuscript, several issues remain insufficiently resolved, particularly regarding the interpretation of findings given the limited sample size. The study includes only 10 participants per group, which substantially limits its statistical power and raises concerns about type I and II errors. While the authors cite similar sample sizes in previous histological studies, this does not obviate the need for cautious interpretation and restrained conclusions. The revised text should consistently reflect the exploratory nature of the work, clearly delineating hypothesis-generating observations from confirmatory evidence. Statements suggesting mechanistic involvement or clinical implications should be further tempered, especially in the abstract, discussion, and conclusion.
Moreover, while the authors have added confidence intervals and effect size estimates for some key variables, statistical reporting remains inconsistent across figures and tables. The newly added confidence intervals are appreciated, but not all correlation findings or group comparisons are accompanied by effect size metrics. This undermines interpretability, particularly in the absence of multivariable adjustments, which the authors justifiably avoided due to underpowered sample size. Nevertheless, the limitations section could further emphasize how this impacts the robustness and generalizability of the observed associations.
Another concern relates to the visual data presentation. Although the authors have improved figure legends and added qualitative descriptions, the immunohistochemistry images remain largely illustrative without rigorous quantification or clear region-of-interest standardization. Semi-quantitative grading and mast cell counts are mentioned, but details on how reproducibility was maintained—especially across multiple observers—are somewhat limited despite the mention of kappa coefficients and Bland-Altman analyses. Including inter-rater agreement statistics in the Results section (not just Methods) and providing supplementary data on scoring reproducibility would greatly enhance transparency.
Lastly, while the mechanistic discussion has been revised to acknowledge its speculative nature, several passages still overstate the role of mast cells in mediating airway remodeling and vascular alterations in COPD-B patients. Without direct functional or molecular evidence, such as mediator profiling or co-localization with angiogenic markers, these interpretations should be rephrased as hypotheses rather than conclusions.
In summary, this pilot study explores a compelling histopathological difference between COPD-B and COPD-T groups. However, given the small sample size, lack of healthy controls, and limited statistical power, the authors must adopt a more cautious tone and limit overinterpretation. With appropriate moderation of claims and improved consistency in statistical and methodological reporting, the manuscript could be a valuable contribution to hypothesis generation in the field of environmental COPD phenotypes.
Author Response
We would like to express our sincere gratitude to the reviewer for the insightful and constructive comments. Your observations have been instrumental in refining the manuscript, helping us present our findings with greater balance, precision, and objectivity.

Round 3
Reviewer 2 Report
Comments and Suggestions for Authors
Please let the editor-in-chief make the final decision.